# Identification and Determination of Seven Phenolic Acids in Brazilian Green Propolis by UPLC-ESI-QTOF-MS and HPLC

**DOI:** 10.3390/molecules24091791

**Published:** 2019-05-09

**Authors:** Shengwei Sun, Meijuan Liu, Jian He, Kunping Li, Xuguang Zhang, Guangling Yin

**Affiliations:** 1Science and Technology Centre, By-Health Co. Ltd., No. 3 Kehui 3rd Street, No. 99 Kexue Avenue Central, Science City, Luogang District, Guangzhou 510000, China; ssw0929@163.com (S.S.); liumeijuan@by-health.com (M.L.); 13662340638@139.com (J.H.); zhangxg2@by-health.com (X.Z.); 2School of Pharmacy, Guangdong Pharmaceutical University, Guangzhou 510006, China; kunping_china@gdpu.edu.cn

**Keywords:** Brazilian green propolis, phenolic acids, UPLC-ESI-QTOF-MS, HPLC, quantitation, methodological verification

## Abstract

Brazilian green propolis is a complex mixture of natural compounds that is difficult to analyze and standardize; as a result, controlling its quality is challenging. In this study, we used the positive and negative modes of ultra-performance liquid chromatography coupled with electrospray ionization quadrupole time of flight mass spectrometry in conjunction with high-performance liquid chromatography for the identification and characterization of seven phenolic acid compounds in Brazilian green propolis. The optimal operating conditions for the electrospray ionization source were capillary voltage of 3500 V and drying and sheath gas temperatures of 320 °C and 350 °C, respectively. Drying and sheath gas flows were set to 8 L/min and 11 L/min, respectively. Brazilian green propolis was separated using the HPLC method, with chromatograms for samples and standards measured at 310 nm. UPLC-ESI-QTOF-MS was used to identify the following phenolic compounds: Chlorogenic acid, caffeic acid, isochlorogenic acid A, isochlorogenic acid B, isochlorogenic acid C, caffeic acid phenethyl ester (CAPE), and artepillin C. Using a methodologically validated HPLC method, the seven identified phenolic acids were then quantified among different Brazilian green propolis. Results indicated that there were no significant differences in the content of a given phenolic acid across different Brazilian green propolis samples, owing to the same plant resin sources for each sample. Isochlorogenic acid B had the lowest content (0.08 ± 0.04) across all tested Brazilian green propolis samples, while the artepillin C levels were the highest (2.48 ± 0.94). The total phenolic acid content across Brazilian green propolis samples ranged from 2.14–9.32%. Notably, artepillin C quantification is an important factor in determining the quality index of Brazilian green propolis; importantly, it has potential as a chemical marker for the development of better quality control methods for Brazilian green propolis.

## 1. Introduction

Propolis is a type of fragrant, gelatinous substance obtained by bees collecting the bud secretions and resins of pine trees, poplars, and other plants. After collection, propolis forms from the mixing of these secretions and resins with beeswax and its parotid secretions [1]. Studies have shown that propolis has a wide range of beneficial biological effects, including antibacterial, anti-inflammatory, anti-viral, anti-tumor, and anti-oxidative properties, as well as the ability to regulate blood lipids and blood sugar. As a result, it has gradually become a hot spot in nutrition research [2].

The propolis deriving from Southeastern Brazil is known as green propolis, owing to both its color and the most important botanical source of propolis: *Baccharis dracunculifolia* (Asteraceae) [3,4]. The composition of propolis is complex and may be affected by plant strain and the geographical environment of the collection; in turn, this complexity is closely related to ultimate biological properties. According to the current literature, there are at least 300 compounds of Brazilian green propolis [5]. Within these, phenolic acids (e.g., caffeic, ferulic, p-coumaric, and cinnamic acids) are its main compounds [6,7,8,9]. Recent years have seen increasing research on the pharmacological activity of propolis, which as driven expansion in its market scale. Advances in the identification and characterization of phenolic compounds are expected to provide reliable quality control metrics for Brazilian green propolis. More specifically, characterization of a single phenolic acid obtained from Brazilian green propolis is important for the selection and production of a bee product that has the highest possible levels of health-promoting compounds.

In recent years, global efforts have been made using different analytical methods to characterize phenols in propolis. Among these, high-performance liquid chromatography (HPLC) combined with mass spectrometry (MS), ultraviolet spectroscopy (UV), or photodiode array detection remain the most important analytical methods [10,11,12,13]. liquid chromatography- mass spectrometry (LC-MS) is a powerful method for the analysis of natural compounds. Given their high sensitivity and accuracy, MS analytical methods offer the potential to discover new secondary components that are difficult to obtain using conventional approaches. More detailed structural information can also be obtained by facilitating the use of tandem mass spectrometry (MS/MS), which allows for the identification of unknown compounds; critically, this identification can occur even without reference to standards [14]. For example, 40 kinds of Portuguese propolis ethanol extracts were extensively analyzed using liquid chromatography (LC), in which diode array detection was combined with electrospray ionization tandem mass spectrometry (LC-DAD-ESI-MS) [15]. The polyphenol fraction of propolis was characterized rapidly and qualitatively by chromatographic electrospray ionization tandem mass spectrometry (HPLC-ESI-MS/MS). The most recent method of HPLC-MS technology is that twelve compounds with antioxidant activities were identified in fermented A. dahurica (FAD) by an high-performance liquid chromatography method coupled with photodiode array detection and electro spray ionization ion trap-time of flight mass spectrometry and 2,2-azino-bis-(3-ethylbenzothiazoline-6-sulfonic acid) diammonium salt (HPLC-PDA-Triple-TOF-MS/MS-ABTS) method [16]. In the present study, high-efficiency, ultra-performance liquid chromatography (UPLC) was combined with HPLC. This combined approach was easy for methodological development in the context of research on phenolic acids in propolis. Importantly, there have been few reports thus far regarding either the use of UPLC-ESI-QTOF-MS to identify phenolic compounds in Brazilian green propolis or the use of HPLC to determine the exact content of identified phenolic acid compounds.

Over the past decade, the increased use and demand for propolis in a variety of products has made its effective quality control a pressing issue. In this study, we present the results of extensive research on phenolic compounds obtained from different Brazilian green propolis samples obtained from different manufacturers. These compounds were identified using accurate-mass, UPLC coupled with UPLC-ESI-QTOF-MS in both the positive and negative modes; these compounds were then characterized using HPLC. This two-fold approach was taken in an attempt to establish the Brazilian green propolis phenolic profile and lay the groundwork for its future use as a quality control strategy. Methodological analysis of HPLC was also conducted, with the aim of performing a scientific assessment of the established analytical method.

## 2. Results and Discussion

### 2.1. Identification of Phenolic Acids Compounds in Brazilian Green Propolis

Brazilian green propolis is collected by bees from bean sprouts, tree exudates, and other plant parts and further modified in beehives. This results in an incredibly complex chemical composition of the resulting propolis. After optimization, our UPLC-ESI-QTOF-MS method was successfully used to identify phenolic acids in Brazilian green propolis. The biggest advantage of UPLC was the quick separation of Brazilian green propolis alcohol extracts, which greatly improved the efficiency of the test. The combination of positive and negative ESI modes was chosen as the ionization method. The QTOF-MS detector allowed for more accurate measurements and higher resolution. Characteristic, common peaks were identified by comparing their chromatographic behavior, UV spectra, and MS information either to those of reference compounds or to reference-related studies [17,18]. Thirty-one compounds were isolated from Brazilian green propolis; of these, 10 phenolic compounds were obtained for later LC-MS analysis [19,20]. Total phenol content was quantified spectrophotometrically and 30 phenolic compounds were identified by HPLC-ESI-MS/MS analysis [21]. The pseudo-molecular ions (M + Na)^+^ and (M − H)^−^ of green propolis were detected in both positive and negative ESI mode. These phenolic acid compounds were then separated using the chromatographic conditions indicated in the experimental section.

Chlorogenic acid (1) with (M + Na)^+^ at *m/z* 377 and (M − H)^−^ at *m/z* 353 was eluted after 2.39 min, whereas caffeic acid (2) with (M + Na)^+^ at *m/z* 203 and (M − H)^−^ at *m/z* 179, isochlorogenic acid B (3) with (M + Na)^+^ at *m/z* 539 and (M − H)^−^ at *m/z* 515, isochlorogenic acid A (4) with (M + Na)^+^ at *m/z* 539 and (M − H)^−^ at *m/z* 515, isochlorogenic acid C (5) with (M + Na)^+^ at *m/z* 539 and (M − H)^−^ at *m/z* 549, caffeic acid phenethyl ester (6) with (M + Na)^+^ at *m/z* 307 and (M − H)^−^ at *m/z* 283, artepillin C (7) with (M + Na)^+^ at *m/z* 323 and (M − H)^−^ at *m/z* 299 appeared at 5.16, 7.53, 7.81, 9.82, 25.95 and 33.08 min, respectively. These compounds were identified by their total ion chromatogram (TIC) (as shown in Figure 1) and primary mass spectrum (Figure 2). The data were consistent with previous research [22] and are reported in Table 1.

Brazilian green propolis has become a popular health supplement due to its many biological properties. Characteristically, it has an herbal odor and a unique, irritating taste. Previous work provided the first evidence that artepillin C was the main, pungent ingredient in the ethanol extract of Brazilian green propolis (EEBP). Moreover, that artepillin C potently activated human transient receptor potential ankyrin 1 (TRPA1) channels [23].

In this study, artepillin C was successfully identified and was the same compound that has previously been shown to have a variety of beneficial, biological activities. Notably, three isomers of chlorogenic acid were identified by the UPLC-ESI-QTOF-MS, which has rarely been reported. Thus, these results indicate that this method is useful for identification of the constituents of Brazilian green propolis. It should be noted that Brazilian green propolis contains predominantly phenolic compounds, including flavonoids and phenolic acid as well as its derivatives [24]. There were many unidentified flavonoids in the total ion chromatogram; moreover, the TIC of the methanol extracts obtained from Brazilian green propolis (Figure 1) did not include analysis of its water extracts. In subsequent studies, we will need to further identify these characteristic compounds obtained from Brazilian green propolis.

### 2.2. Determination of Phenolic Acids in Brazilian Green Propolis

After the seven phenolic acids obtained from the Brazilian green propolis were identified from the total ion chromatogram, we further quantified them using HPLC. According to the study carried out by Cuiping Zhang et al., nine phenolic compounds were quantified using HPLC by comparing them with standard substances [22]. The polyphenol fraction in propolis was quantitatively characterized by HPLC-ESI-MS/MS [25]. Using aqueous ethanol along with the addition of the internal standard veratraldehyde, an RP-HPLC procedure for phenolic compounds was developed and 10 compounds were subsequently quantified [26].

Similarly, and as presented here, the main, characteristic peaks were identified by their chromatographic behavior and UV spectra relative to those reference compounds in the HPLC chromatogram (Figure 3). The seven phenolic compounds were separated using the chromatographic conditions indicated in the experimental section of this paper. The content of each phenolic acid component as obtained from the Brazilian green propolis was based on a linear regression equation of the phenolic acid component standard. Each component’s content ratio was calculated based on the relationship between the peak area of the phenolic acid component in the sample and the injection amount. We also conducted a methodological verification to evaluate the scientific nature of our HPLC-mediated, determination method for different phenolic acid compounds found in Brazilian green propolis.

### 2.3. Method Validation

Methodological verification was performed according to International Council for Harmonization (ICH) guidelines [27] and included measurements for: Linearity estimation, system precision and repeatability, and accuracy and stability.

#### 2.3.1. Linearity

The linearity of the method used to identify the aforementioned compounds was determined by analyzing the standards used. Equations for the calibration curves were determined by plotting the relationship between the corrected peak areas (peak area/migration time), ratio of the analysis and the internal standard and the concentration (μg/mL). The concentrations of all compounds obtained from the Brazilian green propolis samples were calculated based on the peak area ratio and data are shown in Table 2. As indicated, all R^2^ values obtained using linear regression analysis were > 0.99. Linear regression equations for the identified compounds are also presented in Table 2, where y is the peak area and x is the concentration.

The UPLC method enables higher efficiency and higher precision detection. In a previous study, the propolis analysis was conducted using HPLC. In general, and in the case of simple, sensitive, and specific reversed-phase high performance liquid chromatography (RP-HPLC), a rapid determination method of salicin was developed and verified to distinguish poplar gum and propolis [28].

#### 2.3.2. System Precision and Repeatability

System accuracy was assessed by repeated injections (*n* = 6) of the standard mixture. The relative standard deviation (RSD)% values of each compound as well as the chromatogram similarity are calculated; all results indicated that instrument variability was sufficiently low at low concentrations. These results showed that the RSD% of each phenolic acid in peak area is less than 2% and similarity of the chromatogram of each standard mixture sample was more than 98%. Collectively, these results indicated that the precision of the instrument met testing requirements.

The repeatability was obtained as the RSD% by analysis of six same green propolis samples of the standard components and instrumental variability, by taking into account the chromatographic peak similarity: such as peak areas and compound retention. The results were considered satisfactory, since the RSD% (*n* = 6) of seven phenolic acids content is less than 2%. Moreover, the similarity of chromatograms from six propolis samples were more than 98%, indicating sound repeatability of our HPLC analysis.

#### 2.3.3. Accuracy

Accuracy was estimated using recovery experiments performed on random Brazilian green propolis. Regarding accuracy and recovery studies, each of the phenolic acids standards was added to the same Brazilian green propolis. Approximately 20% of the analysis used native content obtained before the extraction process. These results are shown in Table 2. The average recovery rate was more than 95%; the acceptable RSD% for all obtained results was less than 5%. It is worth emphasizing that accuracy studies were undertaken for seven phenolic acid compounds. Moreover, the results were considered to be satisfactory for the purposes of our method.

#### 2.3.4. Stability

Regarding the stability of this approach, sample size, pH of the mobile phase, and different solvent brands were mandatory in order to standardize our results. The stability study was implemented by following a “time-by-time” approach, in which the injection time of a sample of the same Brazilian green propolis sample was altered (0, 4, 8, 12 and 24 h) when conducting HPLC. The chromatographic peak similarity at different time points and the RSD% of the seven standards compounds were greater than 98% and less than 2.0%, respectively. Therefore, the developed method was considered to be stable, recognizing its use in different laboratories.

In summary, the validation parameters evaluated in the present study indicated that the HPLC method met the requirements for quality control of our Brazilian green propolis analysis [26].

### 2.4. Data Analysis

The seven identified phenolic acids obtained from Brazilian green propolis were quantified using a validated HPLC method. The contents of each of these phenolic acid compounds are reported in Table 3. Although Brazilian green propolis comes from different parts of Brazil, there was no significant difference in the content of a given phenolic acid across different kinds of Brazilian green propolis samples. This is likely due to these samples coming from the same plant resin sources. Specifically, isochlorogenic acid B had the lowest content (0.08 ± 0.04), while the artepillin C had the highest (2.48 ± 0.94). Artepillin C has been shown to be the main pungent ingredient in Brazilian green propolis [29]; its content as determined here varied from 0–11%, depending on the geographical origin.

The data obtained here were analyzed using a one-way analysis of variance (ANOVA), followed by Tukey’s honestly significant difference (HSD) test. Tukey’s multiple comparisons test was performed for different phenolic acids groups, with results showing that artepillin C had an extremely significant difference in terms of its content in Brazilian green propolis when compared with other compounds (*p* < 0.01; Figure 4). Regarding the isochlorogenic acid C group, a one-way ANOVA also detected an extremely significant difference between its content in Brazilian green propolis and other tested compounds (*p* < 0.01). The data for the different phenolic acids for each group were normally distributed (Figure 4), a requirement necessary to conduct a one-way ANOVA. Using our validated HPLC method, these results suggest that artepillin C quantification can be used as an important factor for determining Brazilian green propolis quality. Moreover, it has potential for use as a chemical marker for the future development of better quality control measures for Brazilian green propolis.

## 3. Experimental Section

### 3.1. Reagents and Chemicals

Chlorogenic acid, caffeic acid, isochlorogenic acid A, isochlorogenic acid B, isochlorogenic acid C, caffeic acid phenethyl ester (all used as standard) were from Chengdu Alfa Biotechnology Co., Ltd. (Chengdu, Sichuan, China). Artepillin C (used as standard) was purchased from Richmond, VA (USA). Methanol (HPLC grade) and ethanol (95%) were from Merck Chemicals. Formic acid (95%) was from Sigma-Aldrich (Milan, Italy).

### 3.2. Sample Preparation

All Brazilian green propolis samples were collected in summer 2017 from different manufacturers (Figure 5) and stored at −25 °C until analysis [30]. Three samples were randomly selected from 14 samples of green propolis tested, and only 0.1 g of each was weighed into a 10 mL volumetric flask and well mixed. After adding methanol to the propolis sample and diluting to volume (10 mg/mL), it was ultrasonically extracted (360 W, 25 KHz) for 30 min, and then the supernatant was prepared. Resulting supernatants were then filtered using a 0.45 μm filter from VWR (Radnor, PA, USA) prior to detection by UPLC-ESI-QTOF-MS (Agilent Technologies, Waldbronn, Germany).

The standard Chlorogenic acid, caffeic acid, isochlorogenic acid A, isochlorogenic acid B, isochlorogenic acid C, caffeic acid phenethyl ester, Artepillin C (5 mg of each) were placed in a 5 mL volumetric flask and dissolved in methanol. The standard was identified after the analysis of UPLC-ESI-QTOF-MS. The standard solution was stored at 4 °C until later use. The three mL of 95% ethanol was added to 100 mg of each Brazilian green propolis sample. The sample was then extracted using ultrasound (360 W, 25 KHz) for 1 h, after which it was centrifuged at 3000 r/min for 10 min. Finally, the 0.1 mL supernatant was obtained and adjusted to 1 mL with ethanol, after which it was filtered using a 0.45 μm filter. The resulting solution was then used for HPLC.

### 3.3. UPLC-ESI-QTOF-MS

Brazilian green propolis was identified using an Agilent 1290 ESI-QTOF-MS spectrometer and an Agilent 6545 Series UPLC system (Agilent Technologies, Waldbronn, Germany). The instrument was equipped with an electrospray ionization (ESI) source (Dual AJS ESI) and a proprietary Agilent jet stream dual nebulizer. In the UPLC analysis, chromatographic separations of Brazilian green propolis were conducted using a C18 Agilent SB column (Waldbronn, Germany) (RP-18,100 mm × 2.1 mm, 1.8 μm particle size) at a flow rate of 0.3 mL/min with 0.1% formic acid (A) and methanol (B) as solvents (99.9%, HPLC grade; Merck, Darmstadt, Germany). Starting with 25% B, to reach 45% B at 11 min, 55% B at 22 min, 70% B at 29 min, 95% B at 40 min, and then it became isocratic for 4 min. The injection volume was 2 μL for each sample.

The mass spectrometer was operated in the positive and negative electrospray ionization modes using Agilent technology. The optimized ESI operating conditions were as follows: Capillary voltage of 3500 V and drying and sheath gas temperature of 320 °C and 350 °C, respectively. The drying and sheath gas flows were set at 8 L/min and 11 L/min, respectively. The range for mass spectrometry detection was: MS: 100–1700 (*m/z*) and MS/MS: 50–1000 (*m/z*). The ion source temperature was 150 °C. Finally, the nebulizer pressure was 35 psi [31].

### 3.4. HPLC

Quantification of Brazilian green propolis was achieved using different instruments; specifically, a Shimadzu HPLC-20AT (Shimadzu, Japanese) using a Kromasil C18 (AKZONOBEL, Sweden) column (RP-18, 250 mm × 4.6 mm, 5 μm particle size) at a flow rate of 1.0 mL/min. The HPLC system consisted of a binary pump, an auto sampler, and photodiode-array detector, which was software-controlled. The mobile phase used was water/formic acid (999:1, *v*/*v*) (solvent A) and HPLC grade methanol (solvent B) (99.9%, HPLC grade). Elution was performed with a gradient starting with 75% B to reach 30% B at 22 min, 45% B at 45 min, 30% B at 58 min, 20% B at 75 min, 5% B at 80 min, 75% B at 95 min, and then it became isocratic for 10 min. The injection volume for both the sample and the standard was 2 μL. Finally, all chromatograms were measured at 310 nm [32].

### 3.5. Method Validation

In total, 14 samples of Brazilian green propolis were analyzed using HPLC. To verify the rationality of the HPLC method, we conducted a method validation test, which assessed the system’s suitability, linearity, system precision, accuracy, and stability.

#### 3.5.1. Linearity (Calibration Curve)

The calibration curves were constructed using five concentrations—including the Lower Limit of Quantitation (LLOQ)—that ranged from 0.5 to 100 μg/mL. The linear regression equation was determined by taking the sample injection amount as the abscissa and the peak area as the ordinate. The peak area of each standard was considered for the purpose of plotting the linearity graph. Linearity was evaluated using linear regression analysis, which was calculated by the least square regression method.

#### 3.5.2. System Precision

System precision was assessed by repeated injections (*n* = 6) of a standard mixture of the analytes (concentration ≥ Limit of Quantity (LOQ) values). The RSD% values of peak area as well as the similarity of chromatograms were then determined. The acceptance criterion was ±2% for the RSD.

#### 3.5.3. Repeatability

Repeatability was assessed by repeated injection (*n* = 6) of the same Brazilian green propolis sample, weighing approximately 0.25 g. The RSD% values of the seven phenolic acids as well as the similarity of the chromatograms of the six green propolis samples were then determined. The acceptance criterion was ±2% for the RSD.

#### 3.5.4. Accuracy

Accuracy was conducted by a recovery experiment performed on a representative sample (*n* = 6) of Brazilian green propolis. More specifically, equal amounts of standard solution were added to each of the Brazilian green propolis samples and the average recovery rate of the Brazilian green propolis was then calculated. The accuracy method was also evaluated on the basis of RSD.

#### 3.5.5. Stability

The stability of the solution was determined using the green propolis samples for short-term stability by keeping at room temperature for 12 h prior to analysis. Representative propolis samples were then injected at 0, 4, 8, 12 and 24 h. The stability of the instrument detection was judged by the chromatogram similarity at different time points as well as the RSD of the peak area.

### 3.6. Statistical Analysis

The validation experiment as well as the analysis of the quantified sample solution of the Brazilian green propolis extract were performed in six replicates. Experimental data are represented as means ± SD and *p* < 0.05 was considered to be statistically significant. Multiple comparisons were implemented using Tukey’s honestly significant difference (HSD) test. To control for relevant confounding factors, one-way analysis of variance (ANOVA) was used. A *p*-value of < 0.01 indicated extreme statistical significance.

## 4. Conclusions

We present here a simple identification and determination method that was optimized and validated for the analysis of seven phenolic acids. This approach was then applied to the quantitative analysis of 14 Brazilian green propolis samples. When compared with other methods cited in the literature, this approach offers a number of additional advantages, including the use of fewer reagents and reducing the cost of analysis. Given these improvements, one quality control laboratory could analyze more samples per day in addition to saving time and money. And compared with previous publications, the combination of UPLC-ESI-QTOF-MS and HPLC system was the first application in Brazilian green propolis study. The advancement and practicality of the method greatly improved the research of analytical performances. Our results showed that there were some significant differences in phenolic acid content across different Brazilian green propolis samples. Notably, artepillin C has been found only in Brazilian green propolis, indicating promising development for this particular propolis. We report for the first time three different isomers of isochlorogenic acid, indicating the precision and accuracy of our UPLC-ESI-QTOF-MS method. In the present study, we focused on the identification and quantification of different phenolic acids in Brazilian green propolis. Our results suggest that this method could be considered new and effective method that could provide a valuable reference for the future quality control of Brazilian green propolis.

## Figures and Tables

**Figure 1 molecules-24-01791-f001:**
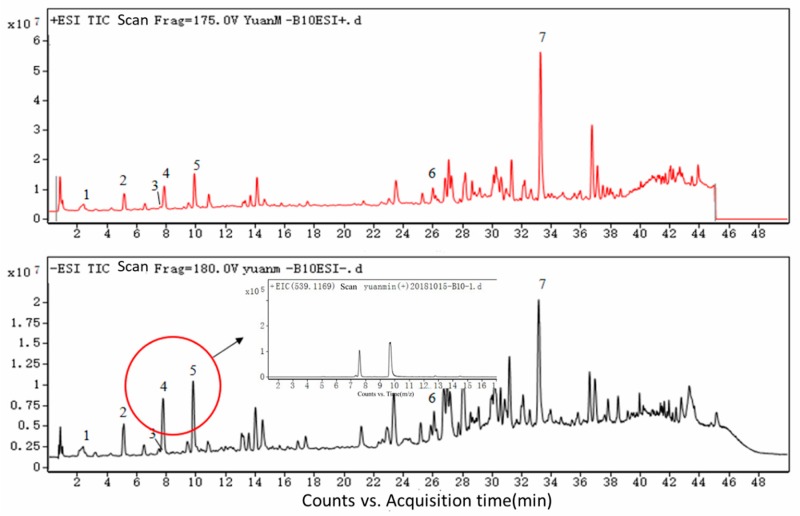
Chlorogenic acid (1), caffeic acid (2), isochlorogenic acid B (3), isochlorogenic acid A (4), isochlorogenic acid C (5), caffeic acid phenethyl ester (6) and artepillin C (7) were identified by ultra-high performance liquid chromatography(UHPLC)-electrospray ionization quadrupole time of flight mass spectrometry (ESI-QTOF-MS), appeared at 2.39, 5.16, 7.53, 7.81, 9.82, 25.95 and 33.08 min, respectively. The absorption of the seven peaks in TIC was similar in both positive and negative ESI modes. However, compound diversity in positive mode was greater than that in negative mode from 24 to 40 min.

**Figure 2 molecules-24-01791-f002:**
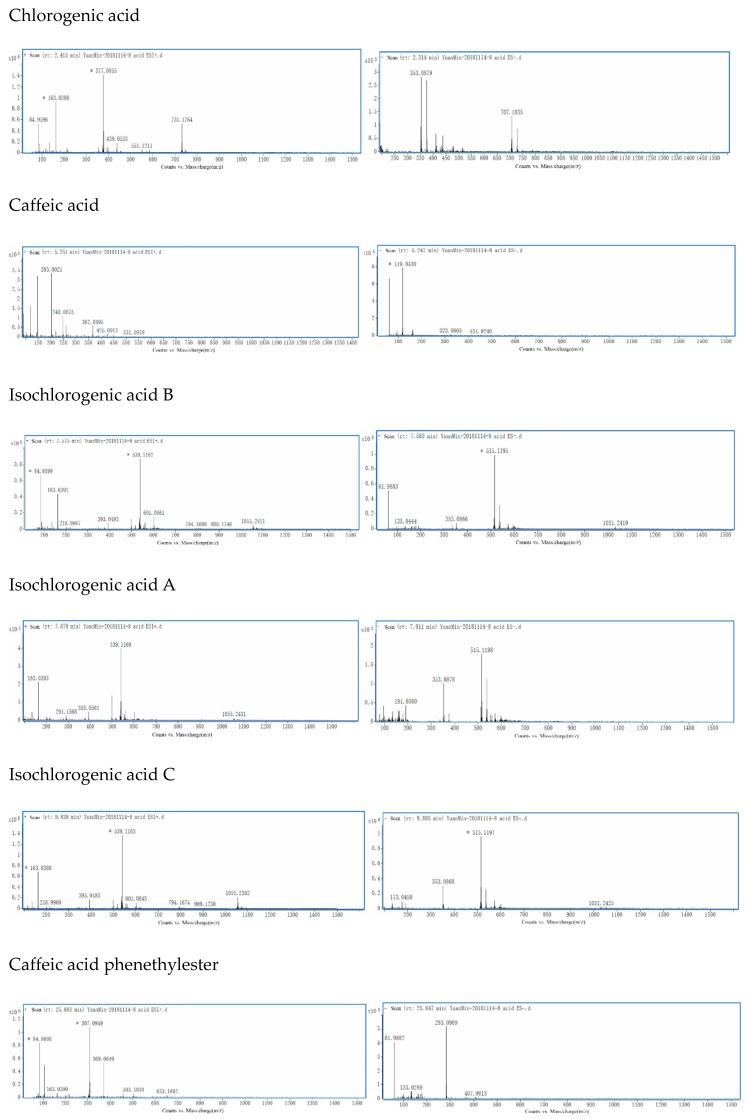
The primary mass spectrum for each of the seven phenolic acids in both positive and negative ESI modes: Chlorogenic acid (1) with (M + Na)^+^ at *m/z* 377 and (M − H)^−^ at *m/z* 353, whereas caffeic acid (2) with (M + Na)^+^ at *m/z* 203 and (M − H)^−^ at *m/z* 179, isochlorogenic acid B (3) with (M + Na)^+^ at *m/z* 539 and (M − H)^−^ at *m/z* 515, isochlorogenic acid A (4) with (M + Na)^+^ at *m/z* 539 and (M − H)^−^ at *m/z* 515, isochlorogenic acid C (5) with (M + Na)^+^ at *m/z* 539 and (M − H)^−^ at *m/z* 549, caffeic acid phenethyl ester (6) with (M + Na)^+^ at *m/z* 307 and (M − H)^−^ at *m/z* 283, and artepillin C (7) with (M + Na)^+^ at *m/z* 323 and (M − H)^−^ at *m/z* 299.

**Figure 3 molecules-24-01791-f003:**
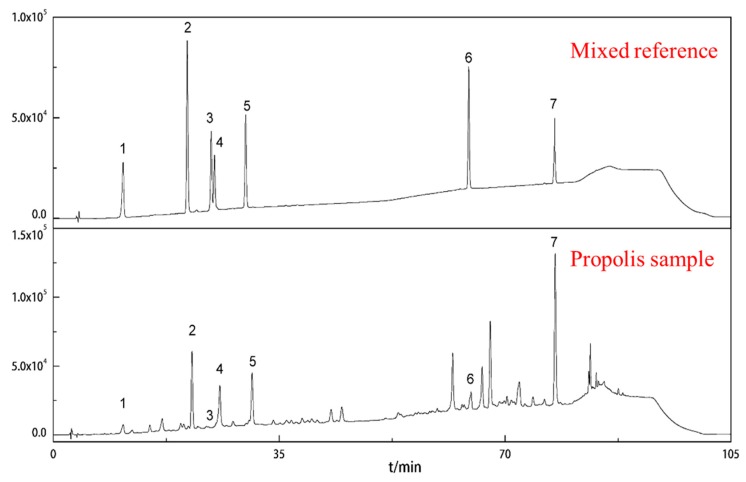
Comparison of chromatograms obtained at 310 nm of chlorogenic acid (1), caffeic acid (2), isochlorogenic acid B (3), isochlorogenic acid A (4), isochlorogenic acid C (5), caffeic acid phenethyl ester (6), and artepillin C (7) mix of standards and Brazilian green propolis sample.

**Figure 4 molecules-24-01791-f004:**
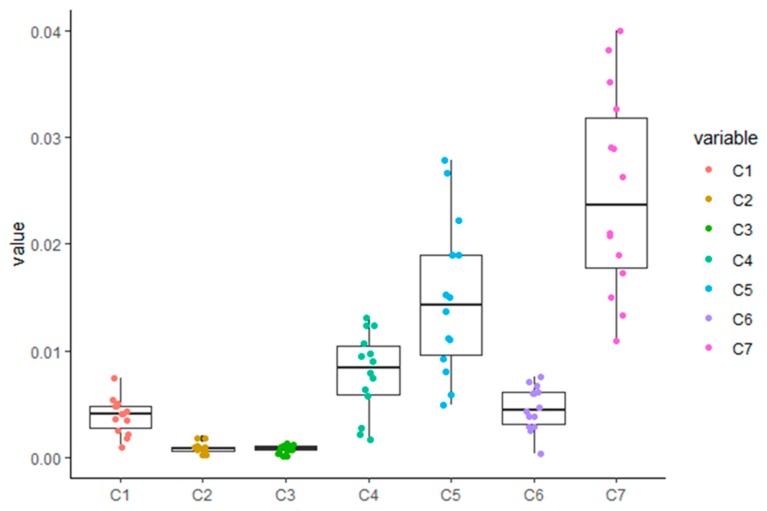
Statistical analysis using one-way (ANOVA) followed by Tukey’s honestly significant difference (HSD) test. C1–C7 are represented with Chlorogenic acid, caffeic acid, isochlorogenic acid B, isochlorogenic acid A, isochlorogenic acid C, caffeic acid phenethyl ester, Artepillin C, respectively. *p* < 0.01 indicates a significant difference between the test groups.

**Figure 5 molecules-24-01791-f005:**
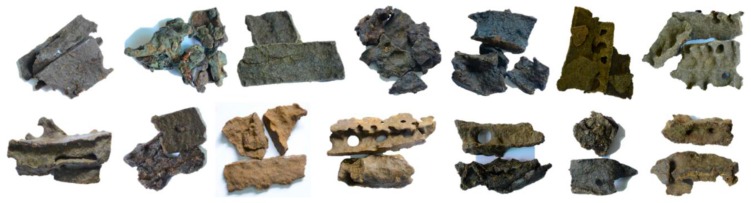
Fourteen Brazilian green propolis obtained from commercial suppliers.

**Table 1 molecules-24-01791-t001:** The identification of seven phenolic acids from Brazilian propolis samples in both positive and negative ESI modes.

No.	t_R_/min	Positive Ion Mode			Negative Ion Mode			Compound
		Molecular Ion (*m/z*)	Molecular Formula	Error (ppm)	Molecular Ion (*m/z*)	Molecular Formula	Error (ppm)	
1	2.391	[M + Na]^+ 377.0855^	C_16_H_18_O_9_	1.591	[M − H]^− 353.0879^	C_16_H_18_O_9_	1.699	Chlorogenic acid
2	5.159	[M + Na]^+ 203.0021^	C_9_H_8_O_4_	−6.206	[M − H]^− 179.0430^	C_9_H_8_O_4_	4.934	Caffeic acid
3	7.526	[M + Na]^+ 539.1169^	C_25_H_24_O_12_	−0.557	[M − H]^− 515.1195^	C_25_H_24_O_12_	0.971	Isochlorogenic acid B
4	7.806	[M + Na]^+ 539.1169^	C_25_H_24_O_12_	0.742	[M − H]^− 515.1198^	C_25_H_24_O_12_	1.553	Isochlorogenic acid A
5	9.820	[M + Na]^+ 539.1169^	C_25_H_24_O_12_	−0.371	[M − H]^− 515.1197^	C_25_H_24_O_12_	1.359	Isochlorogenic acid C
6	25.948	[M + Na]^+ 307.9490^	C_17_H_16_O_4_	0.977	[M − H]^− 283.0969^	C_17_H_16_O_4_	−0.353	caffeic acid phenethyl ester
7	33.078	[M + Na]^+ 323.1618^	C_19_H_24_O_3_	−1.547	[M − H]^− 299.1649^	C_19_H_24_O_3_	0.669	Artepillin C

**Table 2 molecules-24-01791-t002:** Linearity, sensitivity, system precision and accuracy (*n* = 6).

Phenolic Acids	Equation	R^2^	Range (μg/mL)	Average Recovery (Mean ± SD)
^b^ Chlorogenic acid	^a^ Y = 1.25 × 10^6^X − 5405.19	0.9997	3.75–22.50	100.5% ± 3.80%
Caffeic acid	Y = 4.00 × 10^6^X − 8763.30	0.9997	2.70–21.60	96.7% ± 1.39%
Isochlorogenic acid B	Y = 3.09 × 10^6^X − 5324.83	0.9999	0.95–5.70	97.8% ± 2.31%
Isochlorogenic acid A	Y = 2.25 × 10^6^X + 18885.5	0.9999	0.90–11.25	100.4% ± 2.81%
Isochlorogenic acid C	Y = 2.53 × 10^6^X − 55085.10	0.9999	8.50–51.00	99.7% ± 2.51%
caffeic acid phenethyl ester	Y = 2.36 × 10^6^X − 9337.66	0.9996	12.00–72.00	98.5% ± 1.66%
Artepillin C	Y = 3.14 × 10^6^X + 63207.59	0.9987	12.30–98.30	99.5% ± 1.32%

^a^ Response y, is the peak area ratio (area/migration time) of the analytes versus that of standard compound. ^b^ Detection was carried out at 310 nm (Chlorogenic acid, Caffeic acid, Isochlorogenic acid B, Isochlorogenic acid A, Isochlorogenic acid C, caffeic acid phenethyl ester, Artepillin C).

**Table 3 molecules-24-01791-t003:** Determination of seven phenolic acids compounds obtained from fourteen Brazilian green propolis samples.

No.	Compounds (%)							Total Content (%)
	C1	C2	C3	C4	C5	C6	C7	
BP01	0.484	0.095	0.115	1.238	2.662	0.669	2.906	8.17
BP02	0.221	0.111	0.037	0.220	0.588	0.706	1.087	2.97
BP03	0.405	0.095	0.104	0.953	1.525	0.471	3.509	7.06
BP04	0.408	0.181	0.067	0.644	0.917	0.288	1.728	4.23
BP05	0.360	0.055	0.073	0.785	1.117	0.375	1.901	4.67
BP06	0.442	0.070	0.099	1.069	1.899	0.429	2.628	6.64
BP07	0.478	0.093	0.066	0.971	1.901	0.598	3.993	8.10
BP08	0.347	0.080	0.109	0.902	1.501	0.392	2.891	6.22
BP09	0.753	0.178	0.141	0.749	1.370	0.263	1.504	4.96
BP10	0.502	0.097	0.117	1.242	2.780	0.755	3.824	9.32
BP11	0.540	0.094	0.113	1.313	2.232	0.621	3.273	8.19
BP12	0.092	0.017	0.005	0.171	0.494	0.037	1.327	2.14
BP13	0.183	0.024	0.009	0.275	0.797	0.288	2.069	3.65
BP14	0.246	0.060	0.069	0.575	1.110	0.601	2.097	4.76
Mean ± SD	0.39 ± 0.17	0.09 ± 0.05	0.08 ± 0.04	0.79 ± 0.38	1.49 ± 0.72	0.47 ± 0.21	2.48 ± 0.94	5.79 ± 2.2

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
