# Peer review of "Identification and Determination of Seven Phenolic Acids in Brazilian Green Propolis by UPLC-ESI-QTOF-MS and HPLC"

_molecules, 2019, doi:10.3390/molecules24091791_

Round 1
Reviewer 1 Report
The manuscript might have importance to the field, but many studies are already published regarding the chemical composition and activities of green propolis. The manuscript in its present form could not meet the standards for publication in Molecules journal. Extensive English editing must be made, because words are misused, paragraphs with no meaning are presented.
If the authors extensively revise their manuscript, and the Editorial Office of the Journal allows, a future publication may be taken into consideration
Author Response
Thank you for your comments. We showed you the highlights of the manuscript and difference from previous studies,which have been attached in word below.

Reviewer 2 Report
Molecules-487443
Title: Identification and determination of seven phenolic acids in Brasilian green propolis by UPLC-ESI-QTOF-MS and HPLC
This paper presents two analytical methods for the identification and determination of phenolic acids in green propolis samples.
The accuracy and robustness of the obtained results are well established and the number of samples is adequate. The manuscript is well written, and communicates clearly the methodology and results. However, I believe the article should be re-considered after major revisions.
It is not well established on the manuscript why authors use UPLC to identify samples and HPLC to quantify samples. From my opinion the UPLC technology must be also exploited to carry out both the identification and the quantification of the samples.
Authors quantified the phenolic acids in propolis samples by a HPLC method described in bibliography (as reported the MS, reference 25).
In UPLC reported methodology, artepellin (the las peak on the chromatogram) elute at 33 min. and this compound elute around the minute 75 on the HPLC method. So, why use HPLC to quantify compounds?
From my point of view the reported UPLC methodology is of interest to the scientific community. Eliminating the HPLC section and quantifying samples by UPLC method could improve the manuscript.
Drastic changes must be done on the MS to be considered this paper for publication in MOLECULES.
Minor editing changes must be performed.
i.e. improve figure 3. increasing the font size for a better visualization of the mass values
Author Response
Thank you for your kind comments and suggestions. The combination of UPLC-ESI-QTOF-MS and HPLC was used to identification and quantification of phenolic acids in green propolis successfully.The HPLC method was a traditional tool to the analysis of compounds in propolis, but it is friendly to conduct the methodology analysis. The responses of the comments was attached in details.
Thanks again!

Reviewer 3 Report
This manuscript described the analysis of seven phenolic acids in Brazilian green propolis by UPLC-ESI-QTOF-MS. Prenylated derivatives of p-coumaric acid from Baccharis dracunculifolia leaves are the main components of Brazilian propolis. The bioactivities of green propolis may have a relationship with these compounds. However, in this manuscript author only determined the content of artepillin C but not drupanin, capillartemisin A, and the others.
In Figure 4, the HPLC chromatogram of propolis sample, the other peaks near the compounds 6 and 7 should be identified by MS spectra even the standards are difficult to obtain. It can provide more information about the constituents of green propolis. The MS codition can be optimized. Because the phenolic acids (chlorogenic acid and caffeic acid in Figure 3) are easy to identify with ESI(-) MS spectra.
Author Response
Thank you for your comments and suggestions.
The Brazilian green propolis has a complicated chemistry composition, which contains about 300 compounds. Given the standards were used to detect the content of phenolic acids and our target compounds, only seven phenolic acids in Brazilian green propolis were analysed. Moreover, the entire method that contains identification by UPLC-ESI-QTOF-MS and quantification by HPLC can be used to the analysis of green propolis, which can be applied to quality control of green popolis on market. The drupanin, capillartemisin A and other compounds in green propolis may be identified next time with the optimized UPLC-ESI-QTOF-MS method.
In fact, the peaks near the compounds 6 and 7 are some kinds of flavonoid compounds, which have been identified by UPLC-ESI-QTOF-MS. In this manuscript, we focus on the entire method to identify and quantify the phenolic acids in green propolis. About the diversity and bioactivity of compounds in green propolis will be submitted to journal in next time.
More information has been shown in the attachment below.
Thanks again.

Reviewer 4 Report
It is important to point out if the information presented in table 1 was experimentally obtained with the propolis samples or with the standards. In Figure 2, please specify what is the difference between the 2 panels.
It would be important to include an Extracted Ion Chromatogram at 539.1169 to improve the visualization of Isochlorogenic acid isomers.
Figure 3 captions require translation and increase the font size.
In Figure 4, are they chromatograms obtained at ??? nm?
How were you able to quantify compounds 3 and 4 if the resolution was very low?
The resolution observed in the UPLC chromatogram was very low for caffeic acid phenethyl ester, how do the authors confirmed its presence in all the propolis samples?
Author Response
Thank you for your so many kind commets and suggestions.
The responses were attached in details below(in word).
Thanks again, your commets made the article more complete.

Round 2
Reviewer 2 Report
The changes have been well justified by authors, they have answered all the questions that were highlighted in the old version.
The work has been improved, despite of that there still are certain methodological deficiencies, this paper may be of interest to the journal considering the interest of the topic addressed.
Figure 2 does not look good in the attached pdf. Pleas improve the edition of this figure.
Author Response
Thank you for your kind comments. The Figure 2 has been revised. The mass-to-charge(m/z) can be checked clearly after you magnify the picture. The layout of the picture will be typeset according to the requirements of the editorial department.

Reviewer 3 Report
Since the other standards in green propolis are difficult to obtain. The manuscript can be accept in present form.
Author Response
Thanks for your suggestions and comments!
Reviewer 4 Report
Dear authors,
Thank you for considering all the suggestions and answering all the questions.
Very few changes are required:
Figure 1. It is important to point out in the caption:.... positive and negative ESI modes (upper and lower panel, respectively. additionally, the note about "...peak absorptions were higher under the negative ESI mode (e.g., peaks 2, 4, and 5)." is not very clear since the maximum intensity observed in the Y axis is very similar.
Figure 2 still requires changes to improve the quality of the spectra.
Suggestion for Figure 3 caption: Comparison of chromatograms obtained at 310 nm of chlorogenic acid (1), caffeic acid (2), isochlorogenic acid B (3), isochlorogenic acid A (4), isochlorogenic acid C (5), caffeic acid phenethyl ester (6), and artepillin C (7) mix of standards and Brazilian green propolis sample.
Author Response
Thank you for your comments and suggestions.The manuscript has been impoved. More details can be checked in the attachment. And all the revision showed in red words in manuscript.
